# Mitochondrial Dynamics in Non-Small Cell Lung Cancer

**DOI:** 10.3390/cancers16162823

**Published:** 2024-08-12

**Authors:** Agata Dutkowska, Daria Domańska-Senderowska, Karolina H. Czarnecka-Chrebelska, Ewa Pikus, Aleksandra Zielińska, Laura Biskup, Agata Kołodziejska, Paulina Madura, Maria Możdżan, Urszula Załuska, Edward Zheng, Eliza Adamczyk, Konrad Kędzia, Szymon Wcisło, Marcin Wawrzycki, Ewa Brzeziańska-Lasota, Sławomir Jabłoński, Adam Antczak, Michał Poznański

**Affiliations:** 1Department of General and Oncological Pulmonology, Medical University of Lodz, 90-647 Lodz, Poland; agata.dutkowska@umed.lodz.pl (A.D.); aleksandra_zielinska@icloud.com (A.Z.); laura.biskup@student.umed.lodz.pl (L.B.); agata.kolodziejska@student.umed.lodz.pl (A.K.); paulina.madura@student.umed.lodz.pl (P.M.); maria.mozdzan@student.umed.lodz.pl (M.M.); urszula.zaluska@student.umed.lodz.pl (U.Z.); edward.zheng@stud.umed.lodz.pl (E.Z.); adam.antczak@umed.lodz.pl (A.A.); michal.poznanski@umed.lodz.pl (M.P.); 2Department of Biomedicine and Genetics, Medical University of Lodz, 90-647 Lodz, Poland; karolina.czarnecka@umed.lodz.pl (K.H.C.-C.); ewa.pikus@umed.lodz.pl (E.P.); eliza.adamczyk@student.umed.lodz.pl (E.A.); ewa.brzezianska@umed.lodz.pl (E.B.-L.); 3Department of Thoracic, General and Oncological Surgery, Medical University of Lodz, 90-647 Lodz, Poland; konrad.kedzia@umed.lodz.pl (K.K.); szymon.wcislo@umed.lodz.pl (S.W.); marcin.wawrzycki@umed.lodz.pl (M.W.); slawomir.jablonski@umed.lodz.pl (S.J.)

**Keywords:** lung cancer, mitochondrial dynamics, fusion, fission

## Abstract

**Simple Summary:**

Knowledge about the metabolic landscape of cancer cells may provide groundbreaking discoveries in the field of new methods for the diagnosis, prognosis, and treatment of lung cancer. The aim of our study was to assess mitochondrial alterations in the blood of lung cancer patients. We confirmed that fusion and fission protein blood expressions varied between 47 lung cancer patients and 21 healthy people. In the blood of lung cancer patients, fission protein expression is promoted only at an early stage of the disease. In locally advanced and metastatic stages of lung cancer, there is an increase in fusion protein expression. The results of this study provide hope for mitochondrial dynamics understanding in patients with lung cancer, which in the future may contribute to the discovery of new predictive factors for personalized therapy or diagnosis.

**Abstract:**

In lung cancer patients, two complementary abnormalities were found that can cause disruption of the mitochondrial network: increased fusion and impaired fission, manifested by reduced levels of FIS1, a mitochondrial division regulator, and increased expression of MFN1, a mitochondrial fusion mediator. Immunoexpression studies of MFN1 and FIS1 proteins were performed in serum samples obtained from 47 patients with non-small cell lung cancer (NSCLC) and 21 controls. In the NSCLC patients, the immunoexpression of the MFN1 protein was significantly higher, and the FIS1 protein level was significantly lower than in the control group (*p* < 0.01; *p* < 0.001; UMW test). Patients with early, operable lung cancer had significantly lower levels of MFN1 immunoexpression compared to patients with advanced, metastatic lung cancer (*p* < 0.05; UMW test). This suggests that early stages of the disease are characterized by greater fragmentation of damaged mitochondria and apoptosis. In contrast, lower FIS1 protein levels were associated with a worse prognosis. Increased mitochondrial fusion in the blood of lung cancer patients may suggest an increase in protective and repair mechanisms. This opens up questions about why these mechanisms fail in the context of existing advanced cancer disease and is a starting point for further research into why protective mechanisms fail in lung cancer patients.

## 1. Introduction

Non-small cell lung carcinoma (NSCLC) constitutes 80–85% [1] of all lung cancer cases, which is the most common malignancy in the general population, accounting for nearly 2.5 million new cases diagnosed and over 1.8 million deaths in 2022 alone, according to GLOBOCAN [2]. It is notably characterized by a dismal prognosis, with a 5-year survival rate ranging from 10–20% [3].

NSCLC is a heterogeneous group of lung cancers, classified into several subtypes by the WHO, based on histopathological morphology, and immunohistochemical and molecular markers [4]. Nevertheless, the most clinically relevant subtypes of NSCLC are adenocarcinoma (AC), squamous cell carcinoma (SCC), and large cell carcinoma (LCC), which account for 40–50%, 25–30%, and 9–15% of all NSCLC cases, respectively [5,6,7].

NSCLC is a disease among the elderly, as the incidences gradually increase with time and peak at the ages of 65–70 [8]. Although smoking remains the main preventable cause of lung cancer, accounting for 80–90% of cases, the overall decline in tobacco use emphasizes the significance of acknowledging the importance of other significant risk factors, including ambient particulate matter pollution, exposure to asbestos, secondhand smoke, and radon exposure [9,10,11]. Notably, air contamination with radon is associated with a higher incidence of lung cancer among non-smokers [12].

Despite the fact that advanced-stage lung cancer is inherently an uncurable disease with still not yet fully elucidated pathophysiology, the rapid development of molecular biology and our understanding of cellular mechanisms have led to the emergence of theories for the potential pathophysiological processes leading to the malignancy within the lungs. The potential causes of lung cancerogenesis are impaired immunosurveillance and genetic factors, according to one of the oldest theories involving the role of metabolism and its impairment as one of the drivers for cancer development. Most notably, the Warburg effect, first reported in 1924 by Otto Heinrich Warburg, highlights the reliance on glycolysis over 16-fold more efficient oxidative phosphorylation by certain types of cancer cells, despite an aerobic environment [13]. Warburg hypothesized that impaired mitochondria are the underlying cause of cancerous metabolic environment creation. However, the said claim is currently challenged as not every malignant tumor exhibits mitochondria dysfunction, which partially explains the heterogeneity of the NSCLC cell metabolism as both the Warburg effect and oxidative phosphorylation phenotypes are found [14]. Recent studies have significantly expanded our understanding of the metabolic network in cancer cells, providing greater insight into the complexities of biosynthesis, redox homeostasis, and other critical aspects of metabolism that facilitate cell survival and proliferation. Actual evidence presents an alternative view of the classical Warburg effect, which suggests the promotion of glycolysis as a source of intermediates for biosynthesis over provision of acetyl-CoA for the tricarboxylic acid (TCA) cycle. Moreover, certain types of cancer cells are characterized by excessive glutamine consumption to replenish carbon sources in the TCA cycle, providing intermediates for the nucleotide and lipid synthesis crucially needed for rapid proliferation [15,16,17].

Cell metabolism depends on mitochondrial dynamics, which include fission, fusion mitophagy, and biogenesis. Those continuous processes are essential for energy production, cell division, cell differentiation, and cell death. Fusion is the formation of a single mitochondrion by the physical fusion of the outer and inner membranes of two originally distinct and independent mitochondria. Fission, in turn, is characterized by the separation of a single mitochondrion into two or more daughter organelles (Figure 1) [18,19]. It is extremely important to maintain a balance between these processes because they are involved in, among other processes, cell apoptosis and autophagy [20]. Recent studies have indicated a significant association between mitochondrial dynamics and various diseases, such as cancer and inflammation [21]. For example, increased mitochondrial fission has been observed in several human cancer cells, including melanoma, ovarian, breast, lung, thyroid, and glioblastoma cells. Furthermore, some studies have suggested that enhanced mitochondrial fusion may be directly related to the resistance of tumor cells to chemotherapy.

The metabolism and dynamics of mitochondria in rapidly proliferating cancer cells can be altered to increase their survival [22].

Fusion and fission processes are regulated by several specialized proteins. The process of mitochondrial fission is mediated primarily by dynamin-related protein 1 (DRP1) and mitochondrial fission protein 1 (FIS1). The phenomenon of mitochondrial fusion consists of two processes: fusion of the inner mitochondrial membrane, mediated by the protein of human optic atrophy 1 (OPA1), and fusion of the outer membrane, mediated by mitofusin proteins (MFN1 and MFN2) [23]. FIS1, a protein located in the mitochondrial outer membrane, plays a critical role in mitochondrial fragmentation, involving other dynamin superfamily proteins, see Figure 1 [23]. Also, FIS-1, along with endoplasmic reticulum proteins such as BAP31 caspase, promotes apoptosis by inducing mitochondrial fission. This process is enabled by endoplasmic reticulum calcium signals, which lead to the release of cytochrome c to the cytosol and apoptosis progression via Drp1-dependent mitochondrial fission [24,25]. Fis 1 is an important mediator in peroxisomal fission and inhibits the fusion proteins’ GTPase activity, including MFN1, MFN2, and OPA1, but it does not influence the DRP1 activity [26,27].

Mfn1 is a transmembrane protein with a GTPase function, consisting of 742 amino acid residues [28]. Both its N-terminus and C-terminus are oriented toward the cytoplasm. At the N-terminus, there is a GTPase domain responsible for the oligomerization of proteins related to mitochondrial fusion [29]. Mfn1 is also crucial in the activation of OPA1 [30]. 

Both MFN1 and MFN2 are located in the outer mitochondrial membrane and play a crucial role in controlling their fusion through homotypic (MFN1/MFN1 or MFN2/MFN2) or heterotypic (MFN1/MFN2) connections [31].

The downregulation of FIS1 and upregulation of MFN1 were shown to inhibit cell death [32,33]. Hyperfused mitochondria are associated with enhanced oxidative phosphorylation, and during nutrient starvation, they are protected from lysosomal degradation [34]. Moreover, dysfunctional mitochondrial fission directly induces gene instability and centrosome overduplication, which causes cancer cell resistance to apoptotic stimuli [35]. Also, abnormal fusion was reported to be involved in tumorigenesis by promoting the invasion and migration of cancer cells [36]. Numerous studies have noted the downregulation of apoptosis during tumor progression, including lung epithelial cells [27,35].

## 2. Materials and Methods

The study material consisted of serum samples obtained from patients hospitalized in the Department of General and Oncological Pulmonology and the Department of Thoracic Surgery, General and Oncologic Surgery (both from University Teaching Hospital No. 2 in Lodz, Medical University of Lodz, Lodz, Poland) between July 2021 and March 2024.

Patients with confirmed NSCLC treated by either lobectomy, segmentectomy, or pneumonectomy, based on the results of preoperative assessment, were recruited from the Department of Thoracic Surgery, General and Oncologic Surgery. Patients hospitalized due to suspected lung cancer (for lung cancer confirmation) and patients with chronic cough diagnosis were recruited from the Department of General and Oncological Pulmonology. Those patients underwent bronchoscopy or endobronchial ultrasound bronchoscopy (EBUS) for bronchial tree and mediastinal mass assessment. Patients from the latter group, after lung cancer was excluded, were selected for a control group.

The lung cancer group consisted of forty-seven patients with primary NSCLC and no prior chemo- or radiotherapy treatment. Twenty-one patients with no previous cancer history were enrolled in the control group.

The exclusion criteria for this study encompassed individuals with a history of other malignancies, ongoing infectious diseases, or current oncological (potentially mutagenic) treatment. The median age in the patient group was 72 years: 72 for women and 70 for men. Table 1 presents detailed patient characterization, postoperative histopathological verifications of NSCLC samples (according to the WHO Histological Typing of Lung Tumour and IASCLC Staging Project, 7th ed.), and patient smoking status estimated in pack years (PYs). The study was performed following the Helsinki Declaration and the ethical proceedings approved by the Ethical Committee of the Medical University of Lodz, Poland, no. RNN/283/21/KE. All participants provided written informed consent.

### 2.1. Laboratory Procedures 

Biological material was collected, transported, and stored according to reliable protocols. Qualified nurses drew ten milliliters of blood from each participant in the morning or after overnight rest, prior to any planned hospital procedures. Blood samples were used to measure complete blood count (CBC), C-reactive protein (CRP), and NRL (Neutrophil to Lymphocyte Ratio) and those measurements were performed by the analytical hospital laboratory according to the hospital protocols. For serum protein analysis (FIS1, MFN1), blood samples were placed in sterile tubes without an anticoagulant and allowed to clot at room temperature for 30–45 min. After centrifuging at 1000× *g* (2400 rpm) for 10 min, the serum was carefully separated into Eppendorf tubes. The serum samples were stored in pyrogen/endotoxin-free Eppendorf tubes in 250–500 μL aliquots to avoid repeated freeze–thaw cycles and frozen at −80 °C. Serum samples with visible hemolysis were rejected from analysis. In this study, the primary outcome measures were MFN1 and FIS1 protein immunoexpression level, the secondary CRP level, and the Neutrophil to Lymphocyte Ratio.

Serum levels of MFN1 and FIS1 proteins were determined via enzyme-linked immunosorbent assay (ELISA) and by using commercial kits. To determine the level of FIS1, we used EIAb Human FIS1/Mitochondrial fission 1 protein ELISA Kit (Catalog No.13784h, detection range: 0.156–10 ng/mL). To determine the level of MNF1, we used firstly the EIAb Human MFN1/Mitofusin-1 ELISA Kit (Catalog No.15253h, detection range: 0.312–20 ng/mL), but unfortunately for some samples the detection range was insufficient. Our second choice for the MFN1 protein was ELK Biotechnology Human MFN1 (Mitofusin 1) ELISA Kit (Catalog No. ELK5213, detection range: 0.16–10 ng/mL), which was finally used for all the samples. We chose serum dilutions based on the dilution curves we had prepared; i.e., for MFN1 protein detection we diluted the serum 5 times with Sample Dilution Buffer, and for the FIS1 protein we used undiluted serum. We conducted all the analyses in duplicate, and the results were calculated as the mean of two measurements. The final concentration calculation included serum dilution.

To read the results we used BioTek (Agilent Technologies, Santa Clara, CA, USA) Instruments:The ELx800—single-channel reader-assay system, designed to automatically perform endpoint analysis for ELISA-based applications.Gen5^TM^ Microplate Software for Windows.

The results were read within 10 min after adding the stop solution at 450 nm. The Gen5 software requires the standard concentration to be entered so that the sample concentrations and standard curve plotting are performed automatically.

### 2.2. Statistical Analysis

Descriptive statistics (including means, standard deviations, medians, and confidence intervals) were calculated for continuous variables. The Shapiro–Wilk test, performed to assess the data distribution’s normality, showed that for most data the distribution was not normal. That is why the non-parametric tests were carried out: the U Mann–Whitney test, ANOVA Kruskal–Wallis test, and MANOVA test. The inter-group differences in serum protein levels, inflammatory parameters (e.g., CRP, NLR), and anthropometric parameters were examined using the U Mann–Whitney and the Kruskal–Wallis tests. For data with a normal distribution, parametric equivalents were chosen. Associations between serum protein levels and inflammatory parameters were evaluated using Spearman’s correlation coefficients. A *p*-value less than 0.05 was considered statistically significant unless otherwise stated. Statistical analysis was performed using Statistica 13.1 PL (StatSoft, Tulsa, OK, USA).

## 3. Results

Immunoexpression analysis of the studied proteins in serum of patients vs. controls.

In blood serum, in the group of lung cancer patients, the level of MFN1 protein was significantly higher (*p* < 0.01; Mann–Whitney U test; Figure 2) and the FIS1 protein level was significantly lower compared to the healthy control group (*p* < 0.001; Mann–Whitney U test; Figure 2).

Patients with early, operable lung cancer exhibited significantly lower levels of MFN1 protein compared to patients with advanced, metastatic lung cancer, according to the AJCC classification. The differences were statistically significant (*p* < 0.05; Mann–Whitney U test; Figure 3. For the FIS1 protein level, there was no significant relationship with the AJCC classification (*p* > 0.05; Mann–Whitney U test).

We did not find any significant relations between serum FIS1 and MFN1 levels, and the histopathological type of lung cancer (*p* > 0.05; Kruskal–Wallis test), or the cancer staging according to the TNM staging system (the tumor-node-metastasis) (*p* > 0.05; Kruskal–Wallis test, U Mann–Whitney test).

Moreover, there is no correlation between the intensity of inflammation measured by the NLR (Neutrophil to Lymphocyte Ratio) in lung cancer patients and healthy people, or a correlation between the intensity of inflammation and the immunoexpression level of MFN1, FIS1 proteins, and the stage of lung cancer (*p* > 0.05; MANOVA test).

### Protein Expression Analysis in Relation to Patient Age, Gender, Smoking History, and Body Mass Index

Spearman’s rank correlation coefficient revealed a statistically significant positive correlation between the MFN1 level and PY (rho = 0.31, *p* = 0.017; Spearman’s rank correlation), but no significant correlation between FIS1 and PY (*p* > 0.05, Spearman’s rank correlation).

Statistical analysis showed no significant relation or correlations between the levels of protein immunoexpression and patients’ clinical features (age, gender) (*p* > 0.05, U Mann–Whitney test, Kruskal–Wallis test, and Spearman’s rank correlation). We did not find any significant correlations between FIS1 and MFN1 immunoexpression levels and Body Mass Index (BMI); (*p* > 0.05, Spearman’s rank correlation).

## 4. Discussion

Mitochondrial fusion and fission contribute to morphologic changes and are often dysregulated under pathological conditions, including cancers. The assessment of the expression of proteins regulating mitochondrial dynamics in the blood of lung cancer patients has not been studied so far.

In our study, we present two complementary abnormalities that may cause disruption of the mitochondrial network in non-small lung cancer patients: enhanced fusion and impaired fission. The blood phenotype of cancer patients was associated with a decrease in expression of the mitochondrial fission regulator FIS and an increase in expression of the mitochondrial fusion mediator MFN. These data appear surprising in the context of previous reports pertaining to the assessment of mitochondrial dynamics in lung cancer tissue in comparison to healthy tissue. According to certain authors [37], the phenotype observed in various lung adenocarcinoma cell lines was associated with decreased expression of the mitochondrial fusion mediator MFN-2 and elevated expression of the mitochondrial fission regulator DRP-1.

During our research, we compared patients with early and advanced stages of lung cancer to people without cancer disease. The study group was not free from exposure to tobacco smoke or comorbidities. We tried to ensure that the only factor differentiating the compared populations was the presence of lung cancer. This study design was intended to create a homogeneous group of patients in which it would be possible to eliminate bias caused by other factors that might have played a role in mitochondrial dynamics.

As mitochondrial fusion is associated with anti-apoptotic cell protection, its predominance in the blood of lung cancer patients suggests a probable increase in protective and repair mechanisms in the serum of participants from that group. This raises a question and a starting point for further research regarding the reasons for the protection failure in the context of existing cancer.

In our study, patients with early, operable lung cancer (regardless of the histopathological type) had a significantly lower level of expression of the MFN1 fusion protein compared to patients with advanced, metastatic lung cancer. It is probable that patients with early disease had a greater share of the desired fragmentation of damaged mitochondria and apoptosis. So, what caused the reversal of this dynamic at later stages of the disease? In addition, Liu et al. [38] found that there is an inverse correlation between FIS 1 expression and the clinical stage of lung cancer. They showed that FIS 1 expression was lower in stage I lung adenocarcinoma compared to stage II/III/IV lung adenocarcinoma (analysis performed on the data from The Cancer Genome Atlas).

This leads to the conclusion that perhaps in the early stages of lung cancer, the contribution of fusion and fragmentation is small and remains dormant for some reason. Perhaps there is something that does not sufficiently activate the cells’ protection or does not sufficiently direct impaired cells toward apoptosis. Perhaps the cell defect is not visible (tumor mimicry?). The authors revealed also that patients with low FIS1 expression had significantly better survival rates [38]. Patients from our study in whom lower FIS expression was observed compared to healthy people belonged to the clinically poor prognosis group and, although they were not subjected to survival analysis, they tended to contradict the above observation.

The compared studies [37,38] were conducted on lung cancer cell lines and healthy tissue and this does not allow us to directly address our results. A question arises over whether the activity of mitochondrial dynamics proteins observed in the peripheral blood of patients from our study group corresponds to similar mitochondrial phenomena in tumor tissue.

There are several limitations of our study. First, the use of single-center data limits the generalizability of our findings. Additionally, the limited sample size affects the power of our conclusions. The predictive model was not externally validated with an independent cohort from another institution, which may further restrict its generalizability and robustness. Potential confounding variables that were not considered also could have influenced the results.

## 5. Conclusions

Understanding mitochondrial cross-talk in neoplasia could be a breakthrough for diagnosis and prognosis, and the implementation of potential new therapies in lung cancer. The results of studies concerning metabolic alterations in cancerous cells may provide a promising perspective for future research. Fusion and fission protein blood expression vary between lung cancer patients and healthy people. In the blood of lung cancer patients, fission protein expression is promoted only at an early stage of the disease. In locally advanced and metastatic stages of lung cancer, there is an increase in fusion protein expression. By modulating mitochondrial dynamics and function with various agents or methods, cancer cells may be more susceptible to oxidative stress, hyperthermia, or apoptotic signals than normal cells. Data from the current work provide a starting point for understanding cross-talk in peripheral blood cells concerning mitochondrial dynamics in patients with non-small cell lung cancer. However, more research is needed to understand the complexity of the impact of mitochondrial disorders and their involvement in lung carcinogenesis.

## Figures and Tables

**Figure 1 cancers-16-02823-f001:**
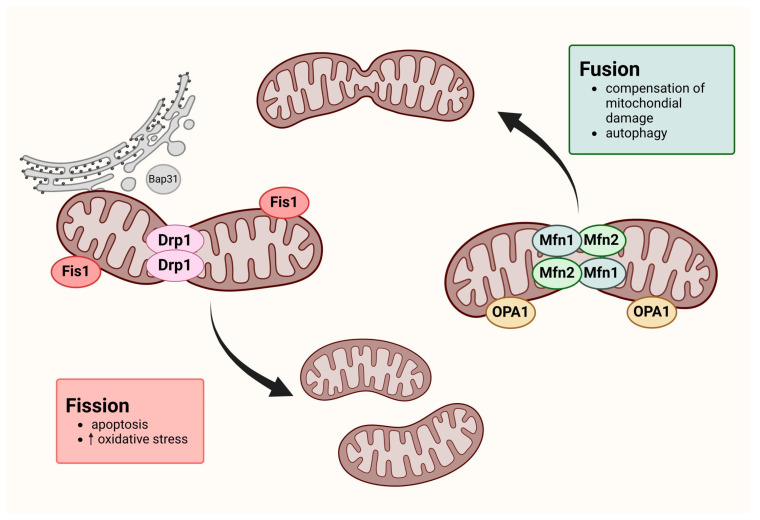
Mitochondrial dynamics—the process of fission and fusion. Created with BioRender.com (figure created by author Mozdzan Maria).

**Figure 2 cancers-16-02823-f002:**
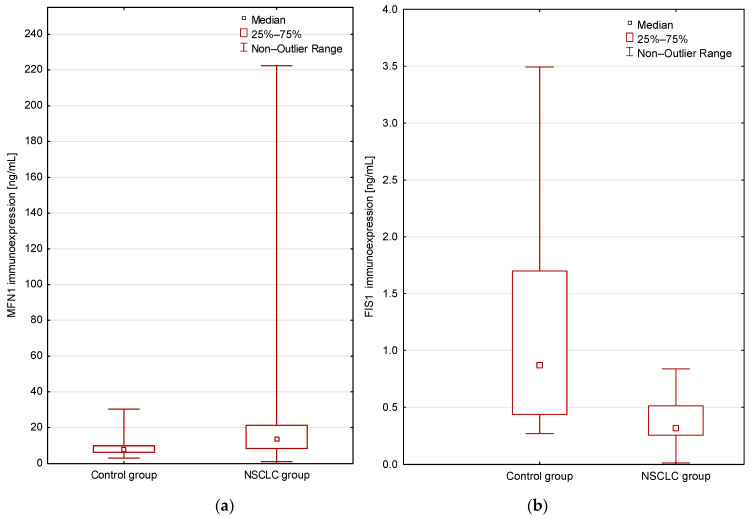
(**a**) Comparison of the MFN1 immunoexpression levels in the study groups (patients vs. controls; *p* = 0.007 Mann–Whitney U test); (**b**) Comparison of the FIS1 immunoexpression levels in the study groups (patients vs. controls; *p* = 0.00006 Mann–Whitney U test). Mean immunoexpression levels of the studied proteins in serum are shown in Table 2.

**Figure 3 cancers-16-02823-f003:**
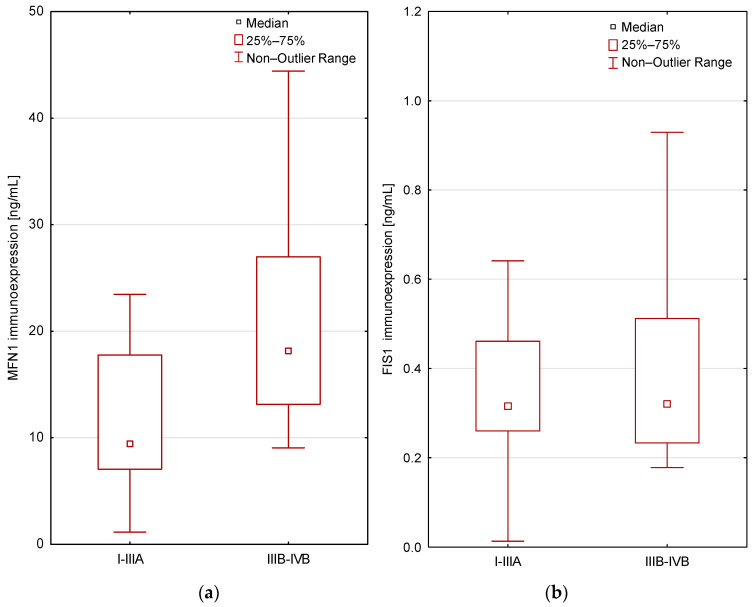
Differences in immunoexpression levels in the study groups according to the AJCC classification (I-IIIA vs. IIIB-IV) for MFN1 (*p* = 0.003; (**a**)), for FIS1 (*p* > 0.05; (**b**)); Mann–Whitney U test.

**Table 1 cancers-16-02823-t001:** Clinical and pathological characteristics of the samples from the study population.

Clinical and Pathological Features	Total
Sample number	68
Control group	21
Patients with lung cancer	47
Median age (years)	72 yrs [IQR: 66–74]
Gender	
Women	35
Men	33
NSCLC cancer subtype	
AC	29
SCC	15
LCC	3
NSCLC cancer characteristics (histopathology)	
TNM scale	
Tumor size	
pT1	5
pT2	14
pT3	11
pT4	16
Node involvement	
N0	18
N1	7
N2	18
N3	3
Cancer metastasis	
M0	33
M1	13
AJCC classification	
AJCC I	6
AJCC II	8
AJCC III	21
AJCC IV	12

**Table 2 cancers-16-02823-t002:** Immunoexpression levels (pg/mL) of MFN1 and FIS1 proteins in patients and controls.

Groups	MFN1	FIS1
Lung cancer patients/NSCLC patients (*n* = 47)	19.8 pg/mL ± 31.5 (IQR: 8.500–21.250)	0.5 pg/mL ± 0.5 (IQR: 0.253–0.512)
Control group (*n* = 21)	10.2 pg/mL ± 6.9 (IQR: 6.400–9.925)	2.2 pg/mL ± 4.4 (IQR: 0.436–1.699)
The significance level	*p* = 0.007	*p* = 0.00006

## Data Availability

All data reported in the study are available in the files of patients kept in our department.

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
