# Peer review of "Mitochondrial Dynamics in Non-Small Cell Lung Cancer"

_cancers, 2024, doi:10.3390/cancers16162823_

Round 1

Reviewer 1 Report

Comments and Suggestions for Authors

In this paper the authors have reported that In lung cancer patients, increased mitochondrial fusion (higher MFN1) and impaired fission (lower FIS1) were observed. ELISA showed significant differences in MFN1 and FIS1 levels between 47 NSCLC patients and 21 controls. Early-stage patients had lower MFN1 levels compared to advanced-stage patients, suggesting greater mitochondrial fragmentation. Lower FIS1 levels were linked to a worse prognosis, highlighting potential protective mechanisms in early disease stages. There are significant flaws in the study and following are my points:

1) Based on a single ELISA test, the authors claim that lung cancer patients exhibit increased mitochondrial fusion (higher MFN1) and impaired fission (lower FIS1). However, the high standard deviations within their sample sizes undermine the conclusiveness of these findings. Given the high variability and reliance on one assay, the conclusions drawn are not sufficiently supported and further validation is required. Furthermore the results did not correlate with the previous TCGA results. 

2) How sure are the authors that the mitochondrial changes seeing in the patient serum are representative of the lung condition? Are there any other supportive studies showing that the patient serum is directly correlated with the mitochondrial state in blood.

3) The standard deviation among the samples is quite high thus can obscure true differences between groups, even when using appropriate tests. The authors did not justify the use if MANOVA as well as did not performed any robustness checks or sensitivity analyses given the high variability in their data. 

4) The discussion lacked the limitations of this study along with the justification of statistical tests used. 

Comments on the Quality of English Language

The English language was ok however there are significant difference in the size of the Font in the paper in a number of paragraphs.

Author Response

We want to thank the Reviewer for his valuable comments and suggestions. Below, we refer to individual points of the received review: 

Reviewer 2 Report

Comments and Suggestions for Authors

A well-written study of NSCLC under the aspect of mitochondrial regulatory and putative prognostic proteins. However, the manuscript is convincing, some minor issues should be addressed:

1. Please unify the representation in Fig. 2 and Fig. 3. The FIS1 group should also be displayed in Fig. 3.

2. Correlation between the intensity of inflammation measured by NLR (Neutrophil to Lymphocyte Ratio) - please include a figure in the main text or the supplements.

3. There are some spelling errors. Please revise the entire text. For example, check the spaces between numbers and units, and before and after the other characters (equal sign etc.).

Comments on the Quality of English Language

Only minor corrections are required.

Author Response

(The authors gave the same response as above.)

Reviewer 3 Report

Comments and Suggestions for Authors

Dear authors,

Whilst an interesting area of research, I feel this manuscript is quite immature for publication as it is. The concept of mitochondrial fission/fusion should be validated in vitro and in publicly available datasets.

Also, before resubmission please ensure font is the same the size through the manuscript.

Author Response

We want to thank the Reviewer for his valuable comments and suggestions. Below, we refer to individual points of the received review.

Whilst an interesting area of research, I feel this manuscript is quite immature for publication as it is. The concept of mitochondrial fission/fusion should be validated in vitro and in publicly available datasets.

Also, before resubmission please ensure font is the same the size through the manuscript.

Response:

Thank you for your comment. Our study is a pilot study that will help initiate further research into mitochondrial disorders in lung cancer. We intend to expand our research with further studies on mitochondrial disorders both in vitro and in vivo. Our study may initiate further research in this area and further analysis of the relationship between the protein and gene expression levels in tissue and serum.

In the current version of the manuscript we ensured that the font is the same the size through the manuscript.

Round 2

Reviewer 1 Report

Comments and Suggestions for Authors

Authors have made the necessary changes.